# Information Rate in Humans during Visuomotor Tracking

**DOI:** 10.3390/e23020228

**Published:** 2021-02-15

**Authors:** Sze-Ying Lam, Alexandre Zénon

**Affiliations:** INCIA, University of Bordeaux, 33076 Bordeaux, France; alexandre.zenon@u-bordeaux.fr

**Keywords:** transfer entropy, information-processing rate, visuo-motor tracking

## Abstract

Previous investigations concluded that the human brain’s information processing rate remains fundamentally constant, irrespective of task demands. However, their conclusion rested in analyses of simple discrete-choice tasks. The present contribution recasts the question of human information rate within the context of visuomotor tasks, which provides a more ecologically relevant arena, albeit a more complex one. We argue that, while predictable aspects of inputs can be encoded virtually free of charge, real-time information transfer should be identified with the processing of surprises. We formalise this intuition by deriving from first principles a decomposition of the total information shared by inputs and outputs into a feedforward, predictive component and a feedback, error-correcting component. We find that the information measured by the feedback component, a proxy for the brain’s information processing rate, scales with the difficulty of the task at hand, in agreement with cost-benefit models of cognitive effort.

## 1. Introduction

Our living environment is rich with stimuli, some of which are crucial in guiding our decisions. Imagine walking into a room full of people: each face, each moving object and each voice in the room are in competition for our cognitive resources. Our brain deals with this overwhelming computational demand by selecting information through attentional mechanisms [1] and by using efficient coding, explaining away predictable data and transmitting only the prediction error, that is the sensory evidence that cannot be predicted from other sources or earlier inputs [2]. According to this view, cognitive resources (e.g., metabolic rate of neurons or information capacity usage) would be dedicated to processing surprising inputs while predictable data would be virtually free to encode [3].

Surprisingly, despite the consensual view of the brain as an information processing machine, few attempts have been made to quantify the amount of information being processed by it, beside the pioneering work described below. One of the reasons for this may be inherent to the technical difficulty of separating predictable from novel information in ecological tasks. The present study aims at filling this gap by applying information theoretic measures to a visuomotor tracking task.

### 1.1. Information Processing Rate in Humans

Just a few years after the publication of Shannon’s seminal paper on information theory [4], several studies attempted to apply this novel theory to estimate the information processing capacity of the human brain. In 1952, Hick [5] compared subjects’ reaction times in a simple forced choice task while varying the number of discrete choices available to them. He observed that the reaction time varied linearly with the logarithm of the number of choices in the task. This result, later coined Hick’s law, implies that there is a constant rate of information gain for this task. This is important because it suggests, counter to intuition, that information processing rate does not vary as a function of task difficulty.

In Hick’s task, the focus was on the decision process and the motor component was assumed constant across conditions. To address this issue, Fitts [6] designed a movement-amplitude control task to estimate information rate with respect to the speed and accuracy of the movement. He first quantified the ‘difficulty’ of reaching a target in information-theoretic terms; then, by dividing this quantity of information by the time it took the subject to attain the target, he obtained an index of performance in units of bits per second, an analogue of information gain in Hick’s task. Fitts found this rate of the human visuomotor-proprioceptive channel to be relatively constant across a range of task conditions (see [7] for a more recent discussion on this matter).

### 1.2. Pursuit-Tracking Task and Its Feedforward Component

In these early attempts at measuring information processing capacity in humans, both Hick and Fitts used simple task designs that involved discrete decisions or movements in each trial. While these might be simpler to study, they do not necessarily provide a good representation of the tasks with which we are faced most often in day-to-day life. To extend the study of human information rate beyond the discrete-task context, Crossman [8] chose to study a pursuit-tracking task. In this experiment, Crossman used an apparatus consisting of a variable-speed velodyne, which drove a piece of paper showing the target course, and a vertical handwheel which subjects used to track it. Importantly, although Crossman’s paradigm involved predictions, a crucial addition for studying information processing during skilled movement [9], the information rate was computed simply as the mutual information between the course and the tracking after correcting for the lag between them, without dissociating the respective contributions of the predictive and error-correcting components. In line with the motor control literature [10,11], these components will be referred to as the feedforward and feedback components respectively. Recent studies have provided evidence in support of the existence of such predictive (feedforward) components during target-tracking in humans. Drop et al. [12] compared three models of tracking on human tracking data and found that the model containing a feedforward component fit their data best. The same authors [13] also tested the effect of the predictability of the target signal on predictive control and found that the degree of reliance on feedforward control is proportional to signal predictability.

To our knowledge, no effort has yet been made to disentangle prediction in a pursuit-tracking task from the more physiologically relevant [14] real-time processing of prediction errors, which we will refer to as the feedback component. Feedforward components, on the other hand, can produce accurate motor responses that are not based on real-time information processing of sensory inputs, but rather on a read-out from the internal model, when faced with predictable data. The present study thus sets out to investigate specifically the role of feedback components of information processing and to leverage the tools of information theory to provide a quantitative description of the real-time information processing rate of human in this visuomotor task.

## 2. Results

Our aim is to study the visuomotor channel that receives visual inputs and generates motor outputs in a one-dimensional visuomotor tracking task with targets of variable predictability (Figure 1).

### 2.1. Background

To give some background to our information-theoretic measures, we start by revisiting the definition of entropy, mutual information, entropy rate, as well as the interpretation of transfer entropy as the rate of information transmission of a channel.

Entropy is the basic quantity we use to measure information. Defined for a random variable *X* with probability p(X), it is given by:(1)$$\begin{array}{c} H(X)=-\sum p(X)\log p(X) \end{array}$$

A channel that takes *X* as input and gives *Y* as output is characterized by a conditional probability function that determines the transition from *X* to *Y*. The rate at which information is processed through such a channel is given by the mutual information between *X* and *Y*:(2)$$\begin{array}{c} \begin{array}{cc} I(X;Y) & =\sum p\left(X,Y\right)\log\frac{p(X,Y)}{p(X)p(Y)}\\ \; & =H(X)-H(X|Y)\\ \; & =H(Y)-H(Y|X) \end{array} \end{array}$$

Mutual information provides insights about the static relationship between two random variables. In order to quantify the dynamics, or causality, of the relationship between multiple random processes, one must consider transition, rather than static, probabilities, which leads to the definition of entropy rate (for a single variable) and transfer entropy (for the interaction of two systems) [15].

Entropy rate measures the rate of growth of entropy of a sequence, that is how much novel information each new sample provides. For a sequence X of *n* random variables, entropy rate is given by:(3)$$\begin{array}{c} \begin{array}{cc} H(X) & =\lim_{n\to \infty }\frac{1}{n}H\left(X_{1},X_{2},...,X_{n}\right)\\ \; & =\lim_{n\to \infty }H\left(X_{n}|X_{n-1},X_{n-2},...,X_{1}\right)\\ \; & \leq H(X_{n}) \end{array} \end{array}$$
when the limits exist. It can be interpreted as the entropy per symbol in the sequence or as the conditional entropy of the last random variable given all the previous ones. For stationary processes, it is proven that both limits exist and that they are equal. Entropy rate is of particular interest to the current study because the continuous visual target movement in the experimental task was constructed as a sequence of target positions presented on the screen. Therefore, the entropy rate of the target position quantifies all the information there is to know about target position, and which could be potentially transferred to tracking response.

The last inequality in Equation (Equation 3) follows from the property of conditioning, which can never increase the entropy of a random variable; the equality is attained for a sequence of independent and identically distributed (i.i.d.) variables, since in that case $X_{n}$ is independent of the preceding symbols and conditioning on them does not reduce the entropy. However, in our study, successive target positions are correlated. We would therefore expect the sequence to have an entropy rate that is smaller than the entropy of the target position, $H\left(X\right)<H\left(X_{n}\right)$. In other words, there is less uncertainty associated with a target position that follows the sequence than one that is randomly drawn at any given time point.

Transfer entropy, representing the information processed with respect to each new element of the input sequence, is defined as the conditional mutual information between the last output and previous inputs, given the history of the outputs:(4)$$\begin{array}{c} I(Y_{n+d};X_{n}^{\left(l\right)}|Y_{n}^{\left(k\right)}) \end{array}$$
where one defines the notation $X_{n}^{\left(l\right)}=\left(X_{n},...X_{n-l+1}\right)$. Parameters *l* and *k* determine the depth of past values one uses to encode the history of *X* and *Y*, respectively, while *d* represents a time difference between *X* and *Y*, assuming the information transfer is not instantaneous. Our analysis, detailed in the next section, allows us to identify specific choices of *d*, *k* and *l* to compute the information transfer from signal *X* to tracking *Y*.

### 2.2. Definition of Measures

**Basic assumptions.** Adopting a model-free approach, we did not make any specific assumptions on the mechanism involved in producing the observed tracking performance. For our analysis, we rely on two key properties of information sharing and transmission in the system.

The first one is an effective time delay. The feedback channel, while engaging in real-time information processing, suffers from a non-reducible time delay in producing motor outputs with respect to the visual input signals. This time delay is referred to as the visuomotor delay (VMD) is a consequence of the physical constraints of the visuomotor system.

The second key property of the system is its autocorrelation. The signals used in the current study were constructed by passing white noise through a sinusoidal filter (order 2). By altering the parameters of the filter we could control the amount of noise that passed through, thus the predictability of the signal. However, regardless of the predictability, the autocorrelation in the target signals was always limited to second order, due to their sinusoidal nature.

**Feedback component information content.** Using the above properties, we were able to fix the free parameters in the transfer entropy formula in Equation (Equation 4), thus tailoring it to the quantification of the information rate of the feedback component, as desired. Given that the expected delay of information transfer from signal *X* to tracking *Y* is the VMD and that the target is a second-order autocorrelated signal, we set the delay *d* between the two processes to be VMD and the depths *l* and *k* to be 2. To ensure independence between successive samples, we further modified the transfer entropy term by conditioning it on $Y_{t-1}$, thus obtaining the following feedback component measure:(5)$$\begin{array}{c} I_{\mathrm{FB}}=I\left(Y_{t};\left\{X_{t-\mathrm{VMD}},X_{t-\mathrm{VMD}-1}\right\}|\left\{Y_{t-\mathrm{VMD}},Y_{t-\mathrm{VMD}-1},Y_{t-1}\right\}\right) \end{array}$$

**Total information and feedforward component.** The total information shared between signal *X* and tracking *Y* has either of two origins: it arises via feedback information transfer with a non-reducible time delay $\left(X_{t-VMD},X_{t-VMD-1}\to Y_{t}\right)$, or it is due to prediction $\left(X_{t},X_{t-1}\to Y_{t}\right)$. This allowed us to compute it as the joint mutual information:(6)$$\begin{array}{c} I_{\mathrm{total}}=I\left(Y_{t};\left\{X_{t},X_{t-1},X_{t-\mathrm{VMD}},X_{t-\mathrm{VMD}-1}\right\}\right) \end{array}$$

This term represents the expected value of total shared information between *X* and *Y*. Deducting the feedback component from it yields the average information attributable to the feedforward component:(7)$$\begin{array}{c} I_{\mathrm{FF}}=I_{\mathrm{total}}-I_{\mathrm{FB}} \end{array}$$

### 2.3. Validation through Model Simulations

In order to validate the quantities derived above, we built a mathematical model of the task based on optimal control theory [16] (see Figure 2A and Method), in which we could manipulate and measure directly the involvement of the feedback component and therefore provide a ‘true’ value against which to compare our $I_{\mathrm{FB}}$ and $I_{\mathrm{FF}}$ measures.

We formulated the visuomotor tracking task as a linear state space model with quadratic regulation cost. We generated the target and joystick dynamics with a set of linear differential equations, which were also included in the transition matrix *A* of the model(i.e., the model had perfect knowledge of the target and joystick dynamics). State representation *s* also included the error between target and joystick coordinates and the regulation objective was to minimise the value of this state element. All state representations were updated at each time step by means of a Kalman filter, on the basis of a novel observation *x* corresponding to the position of the joystick and the target, acquired VMD timesteps before. Optimisation of the control variable *u* was obtained with a model predictive controller, as described in the Methods section.

**Validation of $I_{FB}$ and $I_{FF}$**. To establish the ground truth for the feedback measure, $T_{\mathrm{FB}}$, we took advantage of the linearity of the Kalman filter to directly quantify the information transfer through the feedback pathway in the model by computing the mutual information between observation *x* and state estimates *s* at the Kalman filter level. Given the Gaussian distribution of both the observation *x* and state estimates *s*, the mutual information can be expressed as:(8)$$\begin{array}{c} T_{FB}=I\left(s_{t};x_{t}\right)=\frac{1}{2}*log\left(\frac{|C\Sigma C^{\prime }+R^{\prime }|}{\left|R\right|}\right) \end{array}$$
with Σ being the state covariance matrix, *C* the state-to-observation matrix, and *R* the observation noise.

The feedforward component, on the other hand, is formulated as the mutual information between state estimates at two successive time points *t* and t−1:(9)$$\begin{array}{c} T_{FF}=I\left(s_{t-1};s_{t}\right)=\frac{1}{2}*log\left(\frac{\left|\Sigma \right|}{\left|Q\right|}\right) \end{array}$$
with *Q* being the process noise. $T_{\mathrm{FB}}$ and $T_{\mathrm{FF}}$ were computed for sets of simulation data generated by the model using different predictability levels of input signals, while all other parameters were kept constant. It is important to stress that $T_{\mathrm{FB}}$ and $T_{\mathrm{FF}}$ provide an upper bound on the actual mutual information between inputs and outputs because they are concerned only with state representations at the Kalman filter level and do not take into account the potential loss of information through filtering at the level of the linear quadratic regulator.

We then computed the proposed information theoretic measures $I_{\mathrm{FB}}$ and $I_{\mathrm{FF}}$ on the same data using Gaussian copula estimation (see Methods) and obtained the correlation of the two measures with their respective ground truth values across different predictability levels. Figure 2B shows the high correlation between $T_{\mathrm{FB}}$ and $I_{\mathrm{FB}}$ ($R^{2}$ = 0.999), and that of $T_{\mathrm{FF}}$ and $I_{\mathrm{FF}}$ ($R^{2}$ = 0.999), attesting to the validity of the proposed measures in quantifying component-specific information. 

**Validation of the Estimate for VMD.** While VMD can be directly extracted from the model for simulation data, there is no way to access it directly in real experimental data. We therefore needed to establish an estimate of VMD that could be applied to experimental data. A candidate for such an estimate was the peak latency of the transfer entropy from signal *X* to tracking *Y*, $TE_{X\to Y}$. Since the feedback component is delayed by VMD, the transfer entropy should peak at t-VMD. To evaluate the correspondence of this candidate measure to the true VMD, we generated simulated data corresponding to true VMD values from 9 to 19 frames while all other parameters were kept constant. Notably, the effective VMD of the simulation data was determined by the sum of the visual and motor delay parameters with an additional delay that was inherent to the joystick mechanism and which depended on the parameters of its state space representation (i.e., spring, mass and damping coefficients). Therefore, here again, we were seeking a correlation rather than a strict equality between inferred and reference values. The comparison showed perfect correlation between the peak latency of $I_{\mathrm{FB}}$ and the VMD actually implemented in the model ($R^{2}=1$), validating this estimate of VMD from data.

**Relationship between feedback component and performance lag.** The effect of prediction on tracking performance is two-fold: first, it provides a cognitively efficient way to encode the target signal, thus saving cognitive resources; second, it compensates for VMD by allowing subjects to act in advance, which contributes to a reduction in performance lag (that is, the lag corresponding to maximum cross-correlation between target and tracking). When prediction fails, we would expect the feedback component to take up more information load to maintain performance level. Due to the irreducible VMD of the feedback component, the more it is involved, the more performance lag will tend to VMD. We looked at our simulation data to confirm this effect by observing the relationship between the ratio of performance lag to VMD and the feedback component measure (Figure 2C). We found a strong exponential relationship between the two variables.

### 2.4. Experimental Results

**Identifying VMD from experimental tracking data.** To be able to compute $I_{\mathrm{FB}}$ and $I_{\mathrm{FF}}$, one must first know the VMD of the system. Having confirmed that the peak latency of transfer entropy $TE_{X\to Y}$ corresponds perfectly with VMD in simulation data, we computed the trial-averaged peak latencies of $TE_{X\to Y}$ for each subject from their performance in the most complex condition to obtain an estimate for each subject’s VMD (Figure 3A). Our results showed that VMD lay between 14 and 16 frames (about 230 to 270 ms) for 10 out of the 11 subjects, while one subject showed a VMD of around 380 ms.

**Feedback information rate.** Using subject-specific VMD, we computed the feedback component $I_{\mathrm{FB}}$ using Equation (Equation 5). Results showed that feedback information transfer increased with the complexity of the signal (F(3,40) = 34.9, *p* < 0.0001). Post hoc Tukey HSD test indicated that condition 1 and condition 4 were significantly different from all other conditions while the difference between condition 2 and 3 did not reach significance.

Since $I_{\mathrm{FB}}$ represents information rate per sample, one can obtain subject-specific information processing rates per second by multiplying $I_{\mathrm{FB}}$ by the number of samples in one second. With a frame rate of 60 Hz, we have concluded that the subjects’ real-time information processing rate lies between 1 to 12 bits/s, depending on the complexity of the signal (Figure 3B).

**Feedforward component and predictability of signal.** The feedforward component measure $I_{\mathrm{FF}}$ cannot be interpreted as an information transfer rate per unit of time because, unlike $I_{\mathrm{FB}}$, it is not an independent measure between successive samples. However, it can still be compared across conditions to help us gain insight about the role of prediction with regards to signals of different predictability. We observed a clearly opposite trend relative to that of the feedback component. As predictability of signals decreased, $I_{\mathrm{FF}}$ also decreased, F(3,40) = 30.7, *p* < 0.0001. Post hoc Tukey HSD test once again indicated only condition 1 and condition 4 were significantly different from all other conditions (Figure 3C).

## 3. Discussion

In the current study we have proposed an original information-theoretic approach to evaluate the computational demand of sensorimotor tasks. One of our key contributions was to obtain a decomposition of the total mutual information between inputs and outputs, tailored to dissociate the contribution of real-time processing of prediction errors (referred to as the feedback component) from that attributable to internal predictions (feedforward component).

This approach affords us the opportunity to quantify the information rate of sensorimotor tasks, and hence, to study the information capacity of sensorimotor systems. We hypothesize that the feedback component is a better marker of the amount of cognitive resources required by the task than the total mutual information. Indeed, in a communication channel in which both encoder and decoder are aware of the autocorrelation of the data *X*, predictability can be leveraged to decrease information rate by encoding only the data that is not already predicted by the conditional probability $P(X_{t}|X_{t-1,t-2,...,1},M)$ implemented in the decoder/encoder, achieving entropy encoding, i.e., a code length that is equal to $H(P\left(X_{t}|X_{t-1,t-2,...,1}\right))$ [17]. This part of the data that cannot be predicted corresponds to the feedback component in the present study. In predictive coding models [18], such optimization of encoding through prediction can be understood in terms of firing rate of prediction error neurons. When inputs are perfectly predictable, these neurons would not fire at all, thereby leading to low metabolic costs.

### 3.1. Information Processing Rate in Humans

When applying the discussed measures to a visual tracking task, we found that, whereas the feedforward information increases with predictability of signals, the information rate of the feedback component decreased with predictability.

Our results thus imply that, in our task, information processing rate adapted to signal predictability, in apparent contradiction with Hick’s law. This suggests that the engagement of cognitive resources in the task was balanced against performance goals, in agreement with cost-benefit models of effort [19,20,21]. Participants would thus invest cognitive resources in proportion to their impact on performance. In the case of predictable targets, investing more resources would have only minimal effect on performance, justifying to maintain a low information rate. In contrast, when predictability is low, performance depends more on information rate, explaining larger rates across subjects in this condition. Future studies will determine in more details the nature of this rate-performance trade-off.

### 3.2. Information-Theoretic Approach to Evaluating Tracking Performance

VMD is an important property of subjects’ sensorimotor system, however its direct estimation from tracking data poses some challenges. Ideally, VMD should correspond to the performance lag in a situation where subjects track completely unpredictable signals, i.e., white noise. However, such a signal has too many high frequency components, which subjects are unable to track, making this approach infeasible in practice. We therefore proposed and validated a model-free solution to estimate VMD in tracking tasks. We found VMD values between 230 to 270 ms, in agreement with previous literature on human visuomotor reaction time [22]. We observed sizeable inter-personal differences in VMD but, within subject, VMD varied little across conditions.

A common outcome measure in tracking experiments is the time lag between tracking response and signal [23,24,25,26]. While this lag by itself is a good indicator of performance, normalizing it with respect to VMD highlighted an interesting relationship to the feedback component. In particular, $\log(\frac{\mathrm{PerfLag}}{\mathrm{VMD}})$ has an approximately linear relationship with the FB component. The combination of VMD and real-time information processing rate thus provide a more complete picture of subjects’ performance in a tracking task.

Its model-free nature, coupled with state-of-the-art methods for information estimation, grant our approach enough flexibility for generalizing it to more complex tasks, such as to accommodate higher-dimensional target/tracking spaces [27] or delay-embeddings of random processes [28].

### 3.3. Limitations

A major advantage of the information-theoretic approach is that it is model-free, and thus requires few assumptions. In the present study, we relied only on the following postulates. First, we assumed that the visuomotor system can be viewed as a constant communication channel that takes visual input *X* and gives motor output *Y*, related through the conditional p(Y|X), which is constant over time within subjects and conditions. Second, we assumed that the VMD was constant over time. Were these assumptions incorrect, our measure would still provide a valid average of the actual information rate. Third, the method used to measure mutual information, namely the Gaussian copula, relies on the assumption of normally distributed dependency structure between variables. This makes our estimate a lower-bound to the true mutual information value, since the Gaussian distribution has the highest entropy among all distributions. Fourth, in our measures, for the sake of simplicity, we have conflated the joystick visual input with the motor output. However, in reality, motor noise is added to the motor output such that the joystick position can differ from the intended one [29]. Therefore, the variable *Y* used in our formulae, which corresponds to the joystick cursor position on the screen, does not really represent the motor output but rather motor output corrupted by noise. This simplification leads to a potential underestimation of the true feedback component that is insensitive to variations in motor noise. This should be addressed in future work.

Another limitation of the present work pertains to the resolution of the VMD estimate. Given the discrete nature of the computerized visuomotor tracking task, we can only measure subjects’ VMD up to the resolution that is allowed by the frame rate of the experimental display. With the 60 Hz display system we used in the experiment, the resolution of the VMD is around 17 ms. Future studies can improve the experimental design to allow for higher resolution for studying the tracking performance.

Finally, it is worth mentioning that our proposed measures are tailored to an order-2 target tracking task. A target with a more complex autocorrelation structure would require adapting the mutual information formulae by adding higher-order terms.

## 4. Conclusions

We have presented here a method allowing us to separate feedback and feedforward information rates of a visuomotor tracking task and have shown that both components are influenced by the predictability of signals. We argue that our proposed measure of the feedback component should provide a more relevant measure of task difficulty, cognitive demand and associated metabolic costs than a non-discriminative total information transfer measure. Future studies should aim at comparing this measure with currently existing metrics of cognitive effort and metabolic demands.

## 5. Materials and Methods

### 5.1. Participants

We recruited 11 right-handed subjects (2 males) aged between 20–28 years old from the local university network. They all have normal or corrected to normal vision. All participants have given their consent in written form. The experiment lasted around 1 hour and all subjects were compensated equally for their time.

### 5.2. Experimental Design

The visuomotor task employed for the current experiment was a one-dimensional target tracking task. A vertical bar (3.3 mm wide and 66.5 mm tall) was presented as the visual target on the screen (1024 × 1280) and subjects were asked to follow the movement of the target with a triangular cursor (6.6 mm wide and 13.2 mm tall), which they controlled through a joystick. The target was programmed to move only along the horizontal plane, so subjects were instructed to constrain their joystick movement to this plane during the task, which they could easily achieve by letting the joystick lean on the front end of its pad while moving it sideways. To prevent subjects from cancelling the target movement on the screen by head or eye movements, subjects wre instructed to place their heads on a fixed headstand attached to the table to stabilise their head positions and they are instructed to fixate at a center crosshair during all trials. In addition, we have installed an Eyelink© 1000 + eye tracker (SR Research Ltd., Kanata, ON, Canada) to monitor their eye movements to ensure compliance to the task instruction. The trajectory of the visual target *y* was pre-programmed by passing white noise ϵ through a sinusoidal filter: $a_{1}x_{t}=\epsilon_{t}+a_{2}x_{t-1}+x_{t-2}$, with $a_{2}=-2\sqrt{a_{1}}\cos\left(\frac{\pi }{100}\right)$. We could manipulate the predictability of the signal by altering the parameter $a_{1}$ of the filter controlling the amount of noise that passed through. Figure 1 shows example signal and tracking from condition 1 and 4. Since the target trajectory was pre-programmed, it was completely independent of the subjects’ response during the task. Horizontal joystick movement was registered as the main output response. We further registered vertical joystick movement and discarded trials during which subjects failed to keep to the required plane.

### 5.3. Mutual Information Estimation Using Gaussian Copula

There exist many different methods for estimating mutual information. For the current study, we found the Gaussian copula method to be the most appropriate for our data [30]. Compared to a classic binning or k-nearest-neighbour methods [31] for mutual information estimation, the copula-based method is less subject to sampling bias and it does not require any assumption regarding the distribution of the random variable.

A copula is a multi-dimensional cumulative distribution function (CDF) for which the marginal distributions of all variables are uniformly distributed over the interval [0,1].

For a multivariate random vector $\left(X_{1},X_{2},...,X_{d}\right)$ with continuous CDFs $F_{i}\left(x\right)=P\left(X_{i}\leq x\right)$, one can apply the probability integral transform [32] to obtain uniformly distributed marginals over the interval [0,1]:(10)$$\begin{array}{c} \left(U_{1},U_{2},...,U_{d}\right)=\left(F_{1}\left(X_{1}\right),F_{2}\left(X_{2}\right),...,F_{d}\left(X_{d}\right)\right) \end{array}$$

Using the uniformly distributed marginals, we can define a copula:(11)$$\begin{array}{c} C\left(u_{1},u_{2},...,u_{d}\right)=P\left(U_{1}\leq u_{1},U_{2}\leq u_{2},...,U_{d}\leq u_{d}\right) \end{array}$$

Sklar’s theorem [33] states that every multivariate CDF of a random vector can be expressed in terms of its marginals and a unique copula, if the marginals are continuous.
(12)$$\begin{array}{c} F\left(x_{1},x_{2},...,x_{d}\right)=C\left(F_{1}\left(x_{1}\right),F_{2}\left(x_{2}\right),...,F_{d}\left(x_{d}\right)\right) \end{array}$$

The theorem has the implication that one could separate the dependency structure of a multivariate distribution from its marginals. The copula is the part of the density function that retains all dependencies between variables, and is independent from individual marginal distributions. It was shown that the mutual information between the random variables equals the negative entropy of their corresponding copula [34]. This implies that mutual information, like copula, does not depend on individual marginal distributions but rather depends only on the interaction between variables.

Using the characteristics of the copula and its link to mutual information, we can now estimate MI by computing the corresponding copula density of the random variables. For a faster estimation of the copula entropy, the marginals are transformed to standard Gaussian variables. Since copula entropy is independent of individual marginal distributions, this transformation should not affect the result. However, this transformation will allow the application of the parametric Gaussian model for MI estimation using covariance matrices and joint covariance matrix of the random variables (X,Y).
(13)$$\begin{array}{c} I\left(X;Y\right)=\frac{1}{2}log\left[\frac{|\Sigma_{X}\left|\right|\Sigma_{Y}|}{|\Sigma_{XY}|}\right] \end{array}$$

To obtain a bias-corrected measure, we compute and remove the estimation bias of ln|Σ| using a known analytical solution [30,35]:(14)$$\begin{array}{c} bias=kln2+\sum_{i=1}^{k}\psi \left(\frac{N-i}{2}\right) \end{array}$$
where *k* is the dimensionality of the data and ψ is the digamma function.

### 5.4. Linear-Quadratic Regulator Model

The state space model takes observation vector *x* as input, incorporating both joystick and visual target positions, and models their dynamics through internal state *s*, transition matrix *A*, motor output *u* and control matrix *B*:(15)$$\begin{array}{c} \begin{array}{cc} & s_{t}=As_{t-1}+Bu_{t-1}\\ & x_{t}=Cs_{t} \end{array} \end{array}$$
and we define the cost function to be:(16)$$\begin{array}{c} J=s_{t+N}^{T}Qs_{t+N}+\sum_{k=0}^{N-1}\left(s_{t+k}^{T}Qs_{t+k}+u_{t+k}^{T}Ru_{t+k}\right) \end{array}$$
with *N* being the control horizon. The matrix *A* was composed of the delayed joystick spring-mass system $A_{j}$ and delayed target dynamics $A_{s}$:(17)$$\begin{array}{c} A=\left(\begin{array}{ccccc} A_{s} & \cdots & & & 0\\ & A_{j} & \cdots & & 0\\ 0\cdots 1 & 0\cdots -1 & 0 & \cdots & 0 \end{array}\right) \end{array}$$
with the number of zeros on the last line depending on the visual and motor delays. The transition matrix for the spring-mass system is a 2 × 2 matrix, which is augmented to account for the visual and motor delays:(18)$$\begin{array}{c} A_{j}=\left(\begin{array}{ccccccccc} 0 & 1 & 0 & \cdots \\ 0 & 0 & 1 & \cdots \\ 0 & & & \ddots \\ 0 & & & \cdots & 1 & 0 & \cdots \\ 0 & & & \cdots & 1 & 1 & \cdots \\ 0 & & & \cdots & -0.01 & 0.8 & 1 & \cdots \\ 0 & & & \cdots & & & & \ddots \\ 0 & & & \cdots & & & & & 1\\ 0 & & & \cdots & & & & & 0 \end{array}\right) \end{array}$$
with the number of leading and ending ones depending on the visual and motor delays, respectively. The delayed target dynamics is represented as
(19)$$\begin{array}{c} A_{s}=\left(\begin{array}{cccccc} 0 & 1 & 0 & \cdots \\ 0 & 0 & 1 & \cdots \\ 0 & & & \ddots \\ 0 & & & \cdots & C_{1}^{0} & C_{2}^{0}\\ 0 & & & \cdots & A_{1,1}^{0} & A_{1,2}^{0}\\ 0 & & & \cdots & A_{2,1}^{0} & A_{2,2}^{0} \end{array}\right) \end{array}$$
with *A* and *C* corresponding to the matrices of the actual state space representation of the target signal (Equation (Equation 15)). The control sequence that minimizes the cost function at time *t* is:(20)$$\begin{array}{c} u_{t}=-H_{t}^{-1}F_{t}s_{t} \end{array}$$
where $F_{t}={\hat{A}}^{T}Q\bar{C}$ and $H_{t}$ is the state to observation matrix,
(21)$$\begin{array}{c} H_{t}={\bar{C}}^{T}Q\bar{C}+\bar{R} \end{array}$$

Augmented matrices $\hat{A}$, $\bar{C}$ and $\bar{R}$ were defined as follows:(22)$$\begin{array}{c} \hat{A}=\left(\begin{array}{c} A\\ A^{2}\\ \vdots \\ A^{N} \end{array}\right)\;\bar{C}=\left(\begin{array}{cccccc} B\\ AB & B\\ A^{2}B & & \ddots \\ \vdots \\ A^{N-1}B & & \cdots & & & B \end{array}\right)\;\bar{R}=\left(\begin{array}{ccc} R\\ & \ddots \\ & & R \end{array}\right) \end{array}$$

At every time step, the state vector was updated with new observation data by means of a Kalman filter.

## Figures and Tables

**Figure 1 entropy-23-00228-f001:**
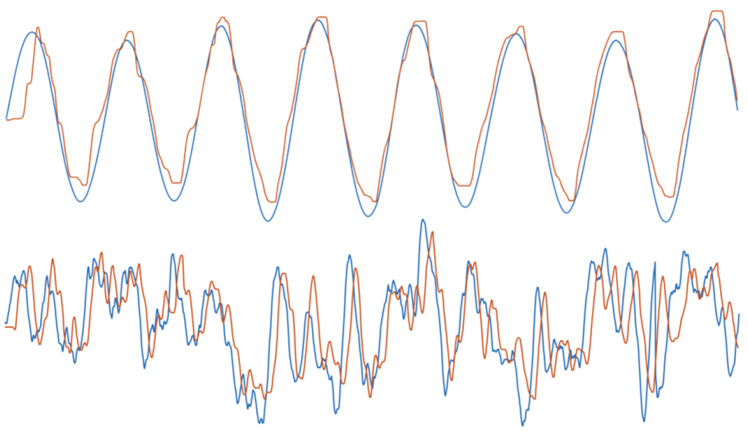
**Example tracking data.** Example experimental data showing the x-coordinates of target signal (blue) and tracking response (orange) for condition 1 (top; most predictable condition) and condition 4 (bottom; least predictable condition).

**Figure 2 entropy-23-00228-f002:**
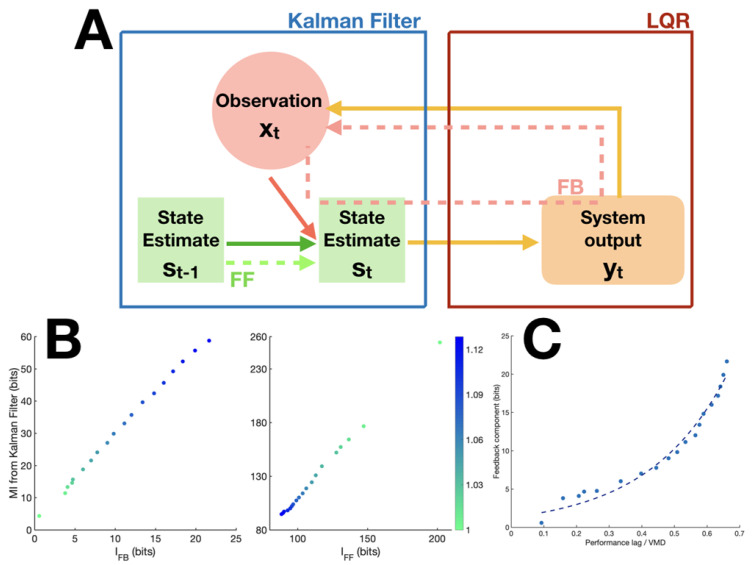
**Simulation design and results.** (**A**) Schematic of Linear Quadratic Regulator model of the visuomotor tracking task. (**B**) Correlation between true feedback measure $T_{\mathrm{FB}}$ and proposed measure $I_{\mathrm{FB}}$ from model data, R = 0.999. (left) Correlation between true feedforward measure $T_{\mathrm{FF}}$ and proposed measure $I_{\mathrm{FF}}$ from model data, R = 0.999. (right) Color code indicates the value of the noise parameter used to generate the signal (see Methods). Larger values correspond to higher complexity in signals, thus less predictable. (**C**) Relationship between $I_{\mathrm{FB}}$ and performance lag/visuomotor delay (VMD) ratio. An exponential function $\frac{\mathrm{PL}}{\mathrm{VMD}}=a\exp\left(bI_{\mathrm{FB}}\right)$ was fitted on the data, with PL the performance lag and VMD the visuo-motor delay. The R squared of the fit was 0.98. a = 1.172 (95% confidence interval: 0.815–1.53), b = 4.282 (3.779–4.785).

**Figure 3 entropy-23-00228-f003:**
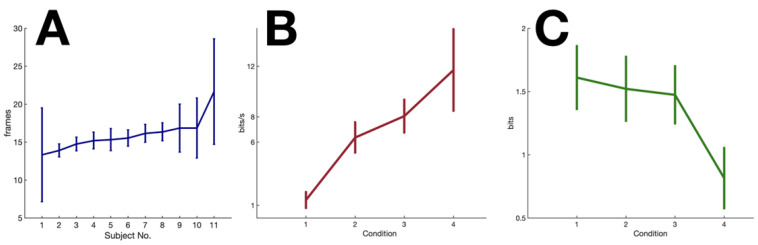
**Experimental results.** (**A**) VMD of individual subjects (sorted in increasing order). Error bars represent the standard deviation across trials. (**B**) Average real time information processing rate per second across subjects (**C**) Average $I_{FF}$ across subjects for different conditions.

## Data Availability

The data presented in this study are openly available in GitHub at https://github.com/szeyinglam/informamtion-rate-tracking, accessed on 8 January 2021.

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
