# Peer review of "Information Rate in Humans during Visuomotor Tracking"

_entropy, 2021, doi:10.3390/e23020228_

Round 1

Reviewer 1 Report

Review of entropy-1087900-v1 " Information rate in humans during visuomotor tracking" by L. Lam & Zenon.

This manuscript reports a study of the information processing rate of the human brain in the context of a visuomotor task (continuous pursuit tracking). The authors use an approach that decomposes the joint mutual information shared between task inputs (observation of target positions and task outputs (behavioral tracking of target position) into a feedforward component dependent on prediction and a feedback component implementing error-correction of incorrect or non-optimal predictions. This decomposition was achieved by representing the joint mutual information as a summation of two modified transfer entropies, one representing the feedforward predictive transfer of information and the other representing the brain’s feedback transfer of information. These two transfer measures were tested on simulated data created via Kalman filtered white noise and empirically-measured human pursuit tracking behavior. The study found that the feedforward and feedback transfer entropy measures performed well against theoretical expectation, with these measures allowing real-time information processing rates reflected in the feedback component to be estimated for the pursuit tracking task. The authors conclude that the latter scales with task difficulty, as expected given cost-benefit models of cognitive effort.

I found this study to be very interesting and important. The authors have provided a novel, model-free way to assess feedforward and feedback human brain information processing that has potential application beyond simple visuomotor pursuit tracking. I believe the proposed measures will be useful for the theoretical and empirical effort to better understand the predictive processing of the brain, as well as applied efforts to study human interaction with technology. From a theoretical standpoint, I find the authors’ reasoning to be persuasive for the most part; from a methodological standpoint the simulations and empirical data collection/analysis were well-implemented. There is much to commend about this study; my only concerns (listed below) involve points of clarification or suggestions to improve the readability of the manuscript.

1) lines 18 – 23, lines 62 – 64, and Lines 243 - 248: Could the authors please clarify their notion of prediction and its association with feedforward processing? Their presentation of this idea seems at odds with how prediction is understood in neuroscientific approaches to predictive processing (or at least how I currently understand it), where prediction is associated with feedback signals within the brain that convey information about an internal model.

2) lines 130 – 133: Here the authors introduce the concept of VMD, but they do not explicitly define it (as time from stimulus to tracking response) until later in the manuscript on lines 143 – 144. I think it would be helpful to state this definition immediately when introducing the concept, as I to re-read the section of the paper a couple of times before understanding what the VMD definition was. My confusion was also due to the statement that “The feedback channel, while engaging in real-time information processing, suffers from a non-reducible time delay in producing outputs with respect to the input signals”. Yet wouldn’t VMD include the feedforward and feedback component time, as the feedforward component is not instantaneous?

3) Lines 145-148: Why does conditioning the transfer entropy on Yt-1 ensure independence between successive samples?

4) Lines 168 – 186: Is the form of Equations 8 and 9 dependent on a certain statistical distribution (Gaussian?) or stems from the structure of the Kalman filter?

5) Lines 156 – 171: I think it would be helpful to improve the introduction of the LQR model. The authors place much of the details of the model in a later methods section, which is acceptable, However, some more of the detail need to be placed in the earlier Results section to assist the reader in understanding the general method/results without having to constantly switch back and forth between the sections. For example, the authors introduce the use of a Kalman filter on line 170 but they do not mention that the filter was used in the LQR model. It is assumed the reader knows this about the model, but readers unfamiliar with this type of model might be confused (as I was on my first reading of this section before I had read the methods). Moreover, the authors mention on lines 134 – 139 that “The signals used in the current study were constructed by passing white noise through a sinusoidal filter (order 2)”. Is this a different filter than that Kalman filter? If they are the same (as seems to be indicated by Figure 2), then the Kalman filter should be introduced in this earlier section (with in-parentheses directions referring the reader to the later sections describing the simulations and the methods). Finally, it seems there is an inconsistency in the acronym of the LQR model (LQR in Fiure 2, but LQG on line 182).

6) Lines 193 – 194: To clarify, does TEX->Y = TFF + TFB?

7) To clarify, computation of the parametric Gaussian model for MI estimation used in the Gaussian copula method (e.g. Equation 13) does not require binning for mutual information computation, correct? All one need do is compute the various correlation matrices from the X and Y time series? My understanding is that even thus parametric model is subject to bias; did the authors include any bias correction terms in this model (Misra, et al., 2005, J. Multivar. Anal. 92, 324–342; Ince et al., 2017, Hum. Brain Mapp. 38, 1541–1573)?

Reviewer 2 Report

In this paper the authors investigate in a visuomotor task the processing of real-time information (as a surprise factor) and the predictable aspects associated with the task. They formalize their findings as two separate components: one that encodes the feedback, error-correcting information and another that captures the feedforward, predictive nature of the task.

For me it has been a pleasure to read the paper. I think the goal of the study is very clear and well supported by the analysis and results. I suggest straightforward acceptance after some small amendments are done.

Some of the references are missing and they appear as question marks. Please check your source files.

Page 10, line 349 wre-> were

Round 2

Reviewer 1 Report

The authors have satisfactorily addressed all of my concerns.